# Multi-Fold Enhancement of Tocopherol Yields Employing High CO_2_ Supplementation and Nitrate Limitation in Native Isolate *Monoraphidium* sp.

**DOI:** 10.3390/cells11081315

**Published:** 2022-04-13

**Authors:** Rabinder Singh, Asha Arumugam Nesamma, Alka Narula, Pannaga Pavan Jutur

**Affiliations:** 1Omics of Algae Group, International Centre for Genetic Engineering and Biotechnology, Aruna Asaf Ali Marg, New Delhi 110067, India; rabinderbiotech@gmail.com (R.S.); asha22@gmail.com (A.A.N.); 2Department of Biotechnology, School of Chemical and Life Sciences, Jamia Hamdard University, New Delhi 110062, India; alka.narula@rediffmail.com; 3DBT-ICGEB Centre for Advanced Bioenergy Research, International Centre for Genetic Engineering and Biotechnology, Aruna Asaf Ali Marg, New Delhi 110067, India

**Keywords:** carbon dioxide, carotenoids, limitation, lipids, microalgae, *Monoraphidium*, nitrate, supplementation, tocopherols

## Abstract

Tocopherols are the highly active form of the antioxidant molecules involved in scavenging of free radicals and protect the cell membranes from reactive oxygen species (ROS). In the present study, we focused on employing carbon supplementation with varying nitrate concentrations to enhance the total tocopherol yields in the native isolate *Monoraphidium* sp. CABeR41. The total tocopherol productivity of NR_HC_ (Nitrate replete + 3% CO_2_) supplemented was (306.14 µg·L^−1^ d^−1^) which was nearly 2.5-fold higher compared to NR_VLC_ (Nitrate replete + 0.03% CO_2_) (60.35 µg·L^−1^ d^−1^). The best tocopherol productivities were obtained in the NL_HC_ (Nitrate limited + 3% CO_2_) supplemented cells (734.38 µg·L^−1^ d^−1^) accompanied by a significant increase in cell biomass (2.65-fold) and total lipids (6.25-fold). Further, global metabolomics using gas chromatography-mass spectrometry (GC-MS) was done in the defined conditions to elucidate the molecular mechanism during tocopherol accumulation. In the present study, the *Monoraphidium* sp. responded to nitrogen limitation by increase in nitrogen assimilation, with significant upregulation in gamma-Aminobutyric acid (GABA). Moreover, the tricarboxylic acid (TCA) cycle upregulation depicted increased availability of carbon skeletons and reducing power, which is leading to increased biomass yields along with the other biocommodities. In conclusion, our study depicts valorization of carbon dioxide as a cost-effective alternative for the enhancement of biomass along with tocopherols and other concomitant products like lipids and carotenoids in the indigenous strain *Monoraphidium* sp., as an industrial potential strain with relevance in nutraceuticals and pharmaceuticals.

## 1. Introduction

The constant growth of the human population requires an increased supply of food and energy, which exacerbates environmental problems such as global climate change. Microalgal cell factories due to their higher growth rates, strong environmental adaptability and metabolic versatility play an essential role in capturing and recycling the global CO_2_, thus feasibly converting it into sustainable biofuels and high-value added biorenewables (HVABs) [1,2,3,4,5]. These photosynthetic organisms excite energy transfer and electron transport within photosystem II (PSII) during light-driven processes resulting in the generation of reactive oxygen species (ROS), which are ineludibly associated with PSII, when absorbed light by the chlorophyll antenna complexes outpaces the rate of energy utilization during CO_2_ fixation [6,7]. Likewise, nitrogen stress and high light (HL) in microalgae leads to increased ROS formation such as singlet oxygen (^1^O_2_), superoxide (O_2_^−^), hydrogen peroxide (H_2_O_2_), and hydroxyl radicals [8,9]. Also, thylakoid membranes are more prone to oxidation due to their higher content of polyunsaturated fatty acids (PUFAs), resulting in the formation of lipid hydroperoxides (LOOHs) and initiating lipid peroxidation chain reactions, thus eventually destroying the chloroplast membrane integrity leading to cell death [10].

Henceforth, these cells have developed multiple mechanisms for treating the inevitable generation of ROS as a by-product of oxidative metabolism. Similarly, the increased accumulation of antioxidant enzymes (superoxide dismutase, glutathione peroxidase, catalase, and ascorbate peroxidase) [11] and other antioxidants like tocopherols and carotenoids tend to scavenge and/or quench ROS in response to oxidative stress [12]. Tocopherols and carotenoids are the major lipid-soluble antioxidants in chloroplast envelope and thylakoid membrane, where photosynthetic light-harvesting and electron transport occur. They play a significant role against photooxidative stress, exhibiting an active defense system contrary to O_2_^−^ and lipid peroxidation in thylakoid membranes [13]. Moreover, tocopherols compensate for the loss of xanthophyll cycle pigments (Zeaxanthin), exhibiting overlapping function during photooxidative stress [14,15,16]. Tocopherols are crucial in delaying the onset of various degenerative diseases in humans and are widely used in dietary supplementation and cosmetics [17]. Furthermore, tocopherols ensure the best utilization of food containing high-quality lipids namely eicosapentaenoic acid (EPA) and docosahexaenoic acid (DHA) to preserve reserves of essential fatty acids [18]. Presently, the industrial applications use chemically synthesized racemic mixtures of tocopherols that are less active than natural molecules (or) directly extracted from vegetable oils that typically contain lower tocopherol yields [e.g., sunflower (900 ug·g^−1^), olive oil (211 ug·g^−1^) and soybean (1.16 mg·g^−1^)] [19,20,21,22,23]. Number of algal strains are also identified as potential producers of tocopherols such as *Euglena gracilis* (2.6 mg·g^−1^), *Nannochloropsis oculata* (1.4 mg·g^−1^), *Coccomyxa* sp. (0.42 mg·g^−1^), and *Haematococcus pluvialis* (0.8 mg·g^−1^) [24,25,26]. In addition, previous studies involving strains such as *Coccomyxa* sp., *Desmodesmus* sp., and *Muriella terrestris* supplemented with 5% CO_2_ (*v*/*v*) demonstrated significant enhancement in the production of both α-tocopherols and total fatty acids [27]. Moreover, the exogenous supply of carbon sources like glucose and ethanol in *Euglena gracilis* have also been found to increase α-tocopherol production [28,29]. Furthermore, screening of nearly 130 strains of microalgae and cyanobacteria by Mudimu and co-workers showed significant α-tocopherol yields at different growth stages, especially under the influence of nitrate limited conditions [30].

In the present study, our focus was to improve tocopherol yields without compromising cell growth, i.e., biomass in response to varying nitrate concentrations subjected to very low (VLC, 0.03% *v*/*v*; 300 ppm) and high (HC, 3.0% *v*/*v*; 30,000 ppm) CO_2_ supplementations in the newly isolated indigenous strain *Monoraphidium* sp. CABeR41. Based on the preliminary data analysis, the overall yield of tocopherols and lipids were induced during nutrient limited (NL) conditions in a well-correlated manner. Furthermore, CO_2_ supplementation drastically enhanced biomass along with other biocommodities such as lipids, carotenoids, and tocopherols. To illustrate the changes in the metabolic profiles of this microalga, we employed qualitative (untargeted) metabolomics to unveil the carbon flux within the essential metabolic pathways. Such findings may provide us new insights on the growth parameters required for improving biomass yields along with the industrially relevant HVABs, a roadmap towards cost-effective and sustainable microalgal biorefineries.

## 2. Materials and Methods

### 2.1. Growth Conditions

Native isolate of freshwater microalgae *Monoraphidium* sp. CABeR41 was grown in minimal BG-11 medium, under 16:8 h light/dark photoperiods with light intensity of ~150 μE m^−2^ s^−1^ at constant shaking of 150 rpm. The final concentration of BG-11 medium used in culture conditions are represented in mg·mL^−1^, mM: K_2_HPO_4_—40, 0.23; MgSO_4_·7H_2_O—75, 0.305; CaCl_2_·2H_2_O—36, 0.21; Citric acid—6, 0.031; FeC_6_H_5_O_7_NH_4_OH—6, 0.022; EDTA—1, 0.0035; Na_2_CO_3_—20, 0.19; NaNO_3_—1.5, 17.6 with trace metal solutions (in mg·mL^−1^, mM): H_3_BO_3_—2.85, 0.045; MnCl_2_·0.4H_2_O—1.81, 0.014; ZnSO_4_·7H_2_O—0.22, 0.001; Na_2_MoO_4_·0.2H_2_O—0.39, 0.001; CuSO_4_·5H_2_O—0.08, 0.0008; Co(NO_3_)_2_·6H_2_O—0.05, 0.005. A total of ~10^6^ cells mL^−1^ was inoculated at a logarithmic phase with an initial optical density at 750 nm wavelength (OD_750_) of 0.1. Growth was monitored by cell count using hemocytometer [31]. The sampling was done at regular intervals of 0, 2, 4, 6, 8, and 10 days. The following equation was used to calculate growth rates [32].
μ = ln (N_2_/N_1_)/(t_2_ − t_1_)(1)
where μ is the specific growth rate and N_1_ and N_2_ are the biomass at times (t_1_ and t_2_), respectively. The following equation was used to compute doubling time:Doubling time = ln (2)/μ(2)

Further, these cells in BG-11 medium were subjected to supplementation of very low carbon [VLC, 0.03% (*v*/*v*) CO_2_ or 300 ppm] and high carbon [HC, 3.0% (*v*/*v*) CO_2_ or 30,000 ppm] with varying concentrations of nitrogen, i.e., nitrogen replete (NR), containing 1.5 g·L^−1^ of NaNO_3_, nitrogen limited (NL) with 0.5 g·L^−1^ of NaNO_3_, and nitrogen deplete (ND) with 0.0 g·L^−1^ of NaNO_3_, respectively. The cells were inoculated in BG-11 medium (as per conditions described earlier) within 500 mL Erlenmeyer flasks and cultivated for period of 10 days at 25 °C under 16:8 h light/dark photoperiods and light intensity of 150 µE m^−2^ s ^−1^ with constant shaking at 150 rpm and the sampling was carried at intervals of 0, 2, 4, 6, 8, and 10 days.

### 2.2. Chlorophyll (Chl_a_) Fluorescence Measurement

Non-invasive fluorescence measurements were acquired by using dual-pulse amplitude modulation (PAM) 100 chlorophyll fluorometer (Heinz Walz Gmbh, Effeltrich, Germany) to measure the photosynthetic efficiency of photosystem II (PSII). To ensure complete oxidation of all reaction centres, samples were kept in dark and incubated for 30 min. Maximum fluorescence (*F_m_*) was determined by directing a saturation pulse (6000 μmol photons m^−2^ s^−1^; λ = 660 nm) and quantum efficiency of PS II calculated by (*F_v_*/*F_m_* = (*F_m_* − *F_o_*)/*F_m_*) [32]. Further, photosynthetic parameters like PSII operating efficiency (*F_q_′*/*F_m_′*) = (*F_m_′* − *F′*)/*F_m_′* [33] and electron transport in PSII (ETRII) were measured as described by Baker and co-workers [34].

### 2.3. Biochemical Profiling

The total lipids were estimation employing the sulpho-phospho-vanillin (SPV) method, wherein 2 mL of cells were pelleted, followed by addition of 2 mL of concentrated H_2_SO_4_ (98%) and incubated at 100 °C for 10 min. After cooling the reaction, 5 mL of freshly prepared phospho-vanillin reagent has been added and incubated at 37 °C for 15 min with continuous shaking at 200 rpm. The absorbance was measured at 530 nm in the SpectraMax M Series Multimode Microplate Reader (Molecular Devices, LLC., San Jose, CA, USA) and the quantification was done using canola oil (MilliporeSigma, Burlington, MA, USA) as the standard [35].

Total carbohydrates were estimated using a modified phenol-sulphuric acid method [36]. For 100 µL of cells, 98% *v*/*v* concentrated H_2_SO_4_ was added and subsequently hydrolyzed at room temperature (RT) for 1 h. Further, 5% (*v*/*v*) phenol with H_2_SO_4_ was added and incubated at RT for 20 min after vortexing. The absorbance was measured at 490 nm and the quantification was done with glucose as a standard.

The total proteins were quantified using a modified biuret method [37]. To, 1 mL of cell pellet, 1 mL of extraction buffer (25% 1 N NaOH in methanol) was added and the reaction was incubated at 80 °C for 15 min. Samples were cooled to RT and centrifuged to remove cell debris. The supernatant was treated with CuSO_4_ solution (0.21% CuSO_4_ in 30% NaOH) and incubated at RT for 10 min before optical density was measured at 310 nm. Quantification was carried by using bovine serum albumin (BSA) as a standard.

### 2.4. Confocal Microscopy with BODIPY Dye

The localization of lipids were visualized using confocal microscopy by the fluorescent dye BODIPY 505/515 (4,4-difluoro-1,3,5,7-tetramethyl-4-bora-3a,4a-diaza-s-indacene; MilliporeSigma, Burlington, MA, USA) as described earlier by Xu et al. [38] with slight modifications. The strained algal cells were incubated in a stock solution of 1 mg mL^−1^ of BODIPY dissolved in dimethyl sulfoxide (DMSO) for 10 min at RT and visualized using Olympus FluoView^TM^ FV1000 Confocal Laser Scanning Microscope (Olympus Corporation, Tokyo, Japan) at an excitation/emission wavelength of 505/515 nm.

### 2.5. Imaging of Superoxide Anion by Confocal Laser-Scanning Microscopy

The superoxide anion imaging was based on the reaction with fluorescent probes such as dihydroethidium (DHE) (MilliporeSigma, Burlington, MA, USA) for detecting the ROS formation and were visualized by the Olympus FluoView^TM^ FV1000 Confocal Laser Scanning Microscope (Olympus Corporation, Tokyo, Japan). To reduce the possibility of artifacts in the fluorescence measurements caused by photooxidation of DHE, the entire procedure was carried out in dark. The stock solution (30 mM) of DHE was prepared by dissolving in 100% (*v*/*v*) methanol and further, the working stock (30 µM) of DHE was diluted in 20% DMSO. Approximately, 10^6^ cells were washed and resuspended 1 mL 1× PBS buffer. Later, 1 µL of 30 µm DHE working stock was added to cell suspension and incubated at RT for 10 min and measurements were done at the excitation/emission wavelength of 520/605 nm as described previously [39].

### 2.6. Quantification of Tocopherols and Carotenoids by High-Performance Liquid Chromatography (HPLC)

The estimation of tocopherols was performed as described by Singh et al. [40] and analyzed in the Agilent HPLC 1260 Infinity II LC System (Agilent Technologies, Santa Clara, CA, USA) using Eclipse Plus C-18 Column (95 Å, 4.6 × 150 mm, 5 µm; Agilent Technologies, Santa Clara, CA, USA) with a two solvent system, i.e., acetonitrile:methanol (60:40, *v*/*v*), at a constant flow rate of 0.6 mL·min^−1^. Further, the detection of tocopherols was done *via.*, fluorescence detector (FLD) at an excitation/emission wavelength of 297/328 nm and quantification was performed using α- and δ-tocopherols as analytical standards obtained from MilleporeSigma (USA).

Extraction of carotenoids in the strain *Monoraphidium* sp. CABeR41 was carried out as reported by Singh et al. [40] with these modifications. After separation of the upper hexane layer, the supernatant was evaporated in presence of nitrogen and the dried extract was reconstituted in methanol. Analysis was carried out in the Agilent HPLC 1260 Infinity II LC System (Agilent Technologies, Santa Clara, CA, USA) coupled with UV detector using C-30 Acclaim column (4.6 × 250 mm, 5 µm) at 35 °C containing a binary solvent system (A plus B) as the mobile phase, with methanol as the primary solvent (A) and methyl tertbutyl ether (MTBE) as the secondary solvent (B). The following gradient conditions were applied (to separate the carotenoids) as follows: 2–20% B for the initial 10 min, followed by 20% B (10–12 min), 20–80% B (12–30 min), 80% B (30–32 min), 80–2% B (32–35 min). All the pigments were identified in UV detector at 437 nm and quantified by comparing the retention times of the each standard obtained from DHI, Denmark [41].

### 2.7. Analysis of Total Antioxidant Activity

The total antioxidant activity were done to evaluate the efficiency of capturing free radicals within the cells in a time-course experiment employing the 2,2-diphenyl-1-picrylhydrazyl (DPPH) [42,43], Total Antioxidant Capacity (TAC) [44], and Ferrous Ion Reduction Power (FRAP) assays [45]. In DPPH assay, to 1 mL of 60 μM methanolic DPPH working stock solution (stock solution was prepared by dissolving 6 mM DPPH in 100% methanol), 10 μL of the extract was added and incubated for 10 min in dark. The absorbance was measured at 517 nm and total antioxidant capacity was calibrated with L-ascorbic acid in terms of ascorbic acid equivalent (mg·g^−1^) [42,43]. For TAC assay [44], 100 µL of the extract was mixed with phosphomolybdate reagent (stock solution containing the following: 1.1 M H_2_SO_4_, 30 mM NaH_2_PO_4_, and 4 mM ammonium heptamolybdate) was incubated at 95 °C for 60 min. The mixture is allowed to cool and absorbance was recorded at 695 nm and activity was measured as described earlier [44]. The reducing power of algal extracts was estimated using Ferrous Ion Reduction Power (FRAP) assay [45]. In this assay, 10 µL of the extract was added to the freshly prepared FRAP reagent containing 5 mL of the TPTZ (2,4,6-tripyridyl-S-triazine) solution (10 mmol·L^−1^) in HCl (40 mmol·L^−1^), 5 mL of FeCl_3_ (20 mmol·L^−1^), and 50 mL of acetate buffer (0.3 mol·L^−1^, pH 3.6), incubated at RT for 10 min. The absorbance was measured at 593 nm with L-ascorbic acid as a standard and represented in ascorbic acid equivalent (mg·g^−1^).

### 2.8. Qualitative (Untargeted) Metabolomics

Approximately, 1 × 10^8^ cells were centrifuged at 8000× *g* for 10 min at 4 °C and immediately quenched in liquid nitrogen; 1 mL of ice-cold methanol/ethanol/chloroform (2:6:2) was added to cells for resuspension, followed by sonication of 15 min in a sonication bath. Samples were centrifuged at 10,000× *g* for 15 min at 4 °C. The supernatant was filtered through a 0.2 µm filter and dried in presence of nitrogen. Further, ribitol (10 mg·mL^−1^) was added as the internal standard and the dried extract was dissolved in freshly prepared methoxyamine hydrochloride solution (4% *w*/*v* in pyridine) and incubated at 30 °C for 90 min. Later, *N*-methyl-N-(trimethylsilyl) trifluoroacetamide was added to solution for derivatization and set for second incubation at 37 °C for 30 min. After centrifugation of the samples for 3 min at 14,000× *g* and the supernatant was further analyzed for different metabolites in the gas chromatography-mass spectrometry (GC–MS/MS). All the sample run conditions along with the instrument setup has been described in detail in our previous reports by Shaikh et al. and Mariam et al. [37,46]. Sample peaks were identified and aligned using the NIST library based on retention time and mass spectral similarity (those hits with R value > 750 were only selected in the study). Further, the final analysis was performed and visualized using MetaboAnalyst 4.0 (https://www.metaboanalyst.ca, accessed on 10 February 2022).

### 2.9. Statistical Analysis

All the experiments were carried out in biological triplicates and are represented as mean average ± SE. Statistical analysis such as two-way Analysis of Variance (ANOVA) with significance of *p*-values < 0.05 and *post hoc* analysis by Tukey’s honestly significant difference (HSD) using agricole package (version 1.3-5).

## 3. Results

### 3.1. Effect of Carbon Supplementation with Varying Nitrate Concentrations on Their Growth Profiles in the Native Isolate Monoraphidium sp. CABeR41

The biomass yields (in g·L^−1^) of the native isolate *Monoraphidium* sp. CABeR41 supplemented with CO_2_ i.e., VLC [0.03% (*v*/*v*) CO_2_ or 300 ppm] and HC [3.0% (*v*/*v*) CO_2_ or 30,000 ppm] in presence of varying nitrate concentrations [nitrogen replete (NR)—1.5 g·L^−1^ of NaNO_3_; nitrogen limited (NL)—0.5 g·L^−1^ of NaNO_3_; and nitrogen deplete (ND)—0.0 g·L^−1^ of NaNO_3_, respectively] is shown in Figure 1 along with nitrate consumption (g·L^−1^) in inset. Our data analysis demonstrates that the biomass productivity of NR_HC_ supplemented cells was significantly higher (*p*-value < 0.05) (336.35 mg·L^−1^ d^−1^), i.e., increased by 4.9-fold with doubling time of 1.60 days than the control (NR_VLC_—56.93 mg·L^−1^ d^−1^ and doubling time of 2.82 days). Similar productivities were observed even in NL_Hc_ supplemented cells (i.e., 316.63 mg·L^−1^ d^−1^ with doubling time of 1.54 days) (*p*-value < 0.05) as on the 10th day of cultivation (Table 1). However, in case of ND_HC_ supplemented cells, the growth was severely hampered (with biomass productivities of 48.19 mg·L^−1^ d^−1^ and doubling time of 6.50 days). In such context, we hypothesize that the presence of additional carbon is diverting the flux towards the cell biomass, i.e., growth along with other biocommodities.

Cellular growth is usually defined by the functioning of organelles such as chloroplast, the photosynthetic apparatus of green cell factories, known to be the primary physiological indicators to illustrate the cell’s perturbation under different environmental conditions [47]. To evaluate the efficiency of photosynthetic apparatus, we have analyzed the PSII reaction centres by measuring the chlorophyll *a* fluorescence. Table 1 represents all the PSII activities when subjected to NR_VLC_, NL_VLC_, ND_VLC_, NR_HC_, NL_HC_, and ND_HC_ conditions in the native isolate *Monoraphidium* sp. CABeR41. Investigating chlorophyll ‘*a*’ fluorescence is a quick and simple indicator to measure stress response in microalgae. Further, the kinetics of chlorophyll fluorescence is useful for monitoring changes in the donor and acceptor sides of PSII reaction centers [48]. As indicated in Table 1, the maximum quantum efficiency of photosystem II (*F_v_*/*F_m_*) in NR_HC_ remained higher at 0.8 on the 10th day of cultivation, which represents intact photoreaction centers and quinone pools that seems to be having better photochemical efficiency in comparison with all other conditions (Appendix A). Moreover, the ETRII represents the rate of non-cyclic electron transfer in the PSII, which was also higher in NR_HC_. However, the NL_HC_ cells recorded (*F_v_*/*F_m_*) 0.7 until the eighth day but a slight decrease was observed at 0.65 on the 10th day, and this may be due to the non-availability of nitrate in medium (Appendix A), whereas in the ND_HC_ condition, *F_v_*/*F_m_* was drastically reduced to 0.24, demonstrating cellular stress with probable damage in the reaction centres (Appendix A). Also, we could not measure photosynthetic activity on the 10th day in the ND_HC_ condition due to complete chlorosis of the cells.

### 3.2. Changes in Biochemical Profiles under the Influence of Carbon Supplementation with Varying Nitrate Concentrations

To understand the effect of carbon supplementation with varying nitrate concentrations in the microalgae *Monoraphidium* sp., we measured the biochemical constituents such as total proteins, total carbohydrates, and total lipids in a time-course experiment for 10 days (Appendix A). Table 2 illustrates the changes in the biochemical profiles of all the macromolecules on the 10th day of cultivation. In the present study, the total protein productivity (197.88 ± 6.5 mg·L^−1^ d^−1^) seems to be highest in NR_HC_ supplemented cells whereas productivities of total carbohydrates (87.12 ± 7.0 mg·L^−1^ d^−1^) and total lipids (76.12 ± 13.4 mg·L^−1^ d^−1^) are highest in NL_HC_ supplemented cells (Table 2), indicating the presence of additional carbon is resulting in diversion of carbon flux towards the enrichment of cellular macromolecules. Also, in the ND_HC_ supplemented cells (Appendix A), the significant increase in the yields of total carbohydrates and total lipids on the 10th day of cultivation is a clear indicator of the stress response in the microalgae and also assumes that the carbon flux is diverting towards these macromolecules with severe compromise in the cellular growth.

### 3.3. Visualization of Lipid Droplets by BODIPY Staining

BODIPY 505/515, a lipophilic bright green fluorescent dye (with several advantages like higher cofficient of molar extraction, strong photochemical ability, and more resistance to photobleaching) has been used to detect intracellular lipids in *Monoraphidium* sp. CABeR41 (Figure 2A). Cells grown in ND_HC_ condition were highly stained, showing the presence of multiple and larger lipid droplets than the other conditions. Also, we have measured the relative fluorescence intensity of the total lipids in the NR_VLC_, NL_VLC_, ND_VLC_, NR_HC_, NL_HC_, and ND_HC_ conditions on the 10th day of cultivation to illustrate the changes occurring within the cells (Figure 2B).

### 3.4. Imaging of Superoxide Anion Acitivity Employing Scanning Microscopy

The dihydroxy ethidium (DHE) fluorescent probes were used to visualise free radical (O_2_^−^) formation. In this reaction, the DHE (low fluorescent) will be first oxidised to DHEox (high fluorescent) in the presence of O_2_^−^ radical and the fluorescence was measured employing confocal laser scanning microscopy. During the non-cyclic electron transport chain (ETC), superoxide radical (O_2_^−^) is formed primarily in the thylakoid-localized PSII, as well as other cellular compartments. Then the superoxide radical (O_2_^−^) initiates lipid peroxidation chain reactions, which eventually destroy the chloroplast membrane integrity and cause cell death during the stress phenomenon. In the Figure 3A,B, the ND_HC_ supplemeted cells demostrated higher DHEox fluorescence followed by ND_VLC_ supplemented cells, which is due to drastic decrease in the quantum efficiency of photosystem II (Fv/Fm) along with the total chlorophyll content (Table 1).

### 3.5. Quantification of Tocopherols and Carotenoids in the Native Isolate Monoraphidium sp. Subjected to Carbon Supplementation with Varying Nitrate Concentrations

Tocopherols are the highly active form of antioxidant molecules involved in interacting with polyunsaturated acyl groups, scavenging of lipid peroxyl radicals, and quenching reactive oxygen species (ROS), thus protecting fatty acids from lipid peroxidation during the stress phenomenon. In the present study, we have quantified two major isoforms of tocopherols in the native isolate *Monoraphidium* sp. CABeR41 subjected to carbon supplementation with varying nitrate concentrations, namely α- and δ-tocopherols. Table 3 demonstrates the significant changes in the total tocopherol productivities observed in NR_HC_ and NL_HC_ supplemented cells than the other conditions on the 10th day of cultivation. Also, Appendix A shows the time-course experiment in the presence of carbon supplementation with varying nitrate concentrations for 10 days. Our data analysis illsutrates that the NL_HC_ supplemented cells resulted in enhancement of 4.06-fold and 8.15-fold in α-tocopherol (244.73 ± 7.44 µg·L^−1^ d^−1^) and δ-tocopherol (489.65 ± 4.53 µg·L^−1^ d^−1^) productivities, respectively, with the total tocopherol productivity at 734.38 ± 11.79 µg·L^−1^ d^−1^ (Table 3, Appendix A). A drastic increase in total tocopherol yields (2743.03 ± 7.21 µg·L^−1^ d^−1^) was observed in ND_HC_ supplemented cells but with impairment of the growth, the overall productivity was decreased.

The quantification of individual carotenoids were estimated when subjected with carbon supplementation with varying nitrate concentrations the native isolate *Monoraphidium* sp. Our data analysis (Table 4) illustrates that NR_HC_ supplemented cells resulted in nearly 0.8-fold increase in total carotenoid yields (2.67 ± 0.15 mg·g^−1^ DCW) in comparison with the NR_VLC_ supplemented cells (1.54 ± 0.08 mg·g^−1^ DCW). However, the primary carotenoids such as α-carotene and β-carotene yields improved in NR_HC_ condition along with a significant increase in zeaxanthin content (1.43 ± 0.07 mg·g^−1^ DCW) (Table 4) and the time-course quantitative analysis of all the carotenoids for 10 days is represented in the supplementary information (Appendix A).

### 3.6. Analysis of Total Antioxidant Activity

The estimation of antioxidant efficiency was performed employing three different methods such as DPPH, TAC, and FRAP assays. Table 5 summarizes the overall activities of these antioxidant assays on the 10th day of cultivation subjected to carbon supplementation with varying nitrate concentrations and also the time-course patterns for 10 days in *Monoraphidium* sp. are illustrated as Appendix A. Our analysis predicts efficient activity was observed in NR_HC_ condition followed by NL_HC_ and other carbon supplemented cells (Appendix A).

### 3.7. Qualitative (Untargeted) Metabolomics

Figure 4A,B and Figure 5, unveil the metabolomic changes occuring in the native isolate *Monoraphidium* sp. CABeR41 subjected to carbon supplementation with varying nitrate concentrations. In the present study, we have observed that the significant increase in biomass was accompanied by enhanced of other commodities such as total tocopherols and total lipids in NL_HC_ supplemented cells which was further subjected to untargeted metabolomics to evaluate the changes resulting in these algal cell factories that will provide new insights for production of industrially relevant molecules such as tocopherols. A total of number of nearly 50 metabolites were analyzed by gas chromtography-mass spectrometry (GC-MS) after filtering the raw data which includes major metabolites namely amino acids, sugars, organic acids, fatty acids, alcohols, antioxidants, and sterols.

Heat map, VIP score, and dot-plot graphs illustrate the overall changes in the metabolome profiles in the NL_HC_ in comparsion with other conditions (Figure 4A,B and Figure 5). A decrease in the relative abundance of amino acids was observed in NL_HC_ condition such as L-alanine, glycine, L-aspartic acid, and L-proline. Whereas, the metabolites like glutamic acid, GABA, trehalose, mannose, and *myo*-inositol were significantly upregulated. Simple sugars, i.e., glucose and galactose, remained unaffected, while sucrose was significantly downregulated. A significant decrease in lactic acid and acetic acid was also observed. Moreover, tricarboxylic acid cycle (TCA) metabolites like malic and fumaric acid are increased, while succinic acid and 2-ketoglutaric acid were decreased. Further, fatty acids like palmitoleic acid, α-linolenic acid, linoleic acid, and steric acid were increased significantly in NL_HC_ supplemented cells. In addition, the other essential metabolites namely phytol and tocopherols were also found to be upregulated (Figure 4A,B and Figure 5). Also, in the Appendix A the linear regression analysis depicting the co-relation between total lipids vs. total tocopherols in native isolate *Monoraphidium* sp. was illustrated in (A) ND_HC_ and (B) NL_HC_.

## 4. Discussion

It is now widely acknowledged that high-value added biorenewables accumulation in microalgae cannot occur without efficient carbon conversion within these major carbon sinks [49]. Harnessing cultivation conditions, efficient utilization of micro- and macronutrients is considered as an useful approach for the production value added bioproducts. Further, based on the specific strain, biomass productivities can be higher with an adequate supply of nutrients such as C, N, P, S, light and CO_2_, but content of lipids and HVABs appears to be lower than expected [50,51,52]. Moreover, there is always metabolic conflict between biomass and storage molecule assimilation during stress, which limits and directs carbon flux either towards biomass accumulation and/or towards lipid and carbohydrate biosynthesis, as a consequence of which one of the components is drastically decreased [53]. Additionally, the integrating of lipid biosynthesis with HVAB yields (i.e., tocopherols, carotenoids etc.) will provide significant additional revenue as a cost-effective recurring bonus for the algal biorefineries.

In the present work, growth and cellular physiology of fresh water microalgae *Monoraphidium* sp. CABeR41 was examined in very low carbon [VLC; 0.03% (*v*/*v*) CO_2_] and high carbon [HC; 3% (*v*/*v*) CO_2_] subjected to varying nitrate concentrations as nitrogen replete (NR; 1.5 g·L^−1^ of NaNO_3_), nitrogen limited (NL; 0.5 g·L^−1^ of NaNO_3_) and nitrate deplete (ND; 0.0 g·L^−1^ of NaNO_3_). The biomass productivity of 3%CO_2_ (NR_HC_) supplemented cells was significantly higher (336.35 mg·L^−1^ d^−1^) with reduced doubling time (1.60 days) compared to the control (NR_VLC_) on the 10th day of cultivation (Table 1). Similarly, NL_HC_ supplemented cells demonstrated higher biomass productivity (316.63 mg·L^−1^ d^−1^) with reduced doubling time (1.54 days). Despite having 0.5 g·L^−1^ of the nitrate than the NR_HC_, the NL_HC_ displayed no apparent phenotypic changes under 3% CO_2_ (*v*/*v*), showing that high CO_2_ supplementation acts as a major contributor in the growth enhancement. Studies in *Chlorella saccharophila* and *Microchloropsis gaditana* also demonstrated that the growth was significantly elevated in 3% CO_2_ supplementation with reduced doubling times [32,54]. Moreover, the growth arrest was observed in the ND_HC_ supplemented cells with lower biomass productivities of 48.19 mg·L^−1^ d^−1^. Further, significant increase in lipid content was observed in ND_HC_ & ND_VLC_ supplemented cells (Table 2), which was shown by BODIPY staining (Figure 2A,B). This might be attributed to the fact that under nitrogen deficiency, the carbon dioxide fixed is diverting its flux into carbohydrates and/or lipids rather than proteins owing to nitrogen deficiency [55]. Another possible reason is that during nitrogen stress, NADPH consumption decreases owing to lack of nitrogen, which inhibits the amino acid biosynthesis, notably the conversion from α-ketoglutarate to glutamate, which results in excess of NADPH within the cells [56]. Also, our data analysis showed a significant increase in carbohydrates and lipid productivities (8.7 and 6.3-fold increase) in NL_HC_ condition. Increase in carbohydrate and lipid productivities under NL_HC_ condition seems to be advantageous indicator for the native isolate *Monoraphidium* sp. CABeR41 as an ideal candidate for CO_2_ sequestration and simultaneous production of energy molecules [57].

The activation of assimilatory pigments, which is centralized by the light gathering antenna complex, is the first committed step in photosynthesis which enables higher rate of CO_2_ uptake and biomass concentration [58]. As shown in Table 1, NR_HC_ supplemented cells depicted increase in total chlorophyll content accompanied by an increase in *F_v_*/*F_m_*, *F_q_′*/*F_m_′*, and ETRII. Moreover, higher *F_v_*/*F_m_* ratio is commonly regarded as increased photosynthetic efficacy, whereas a lower ratio implies PSII photoinhibition [59]. Similar results were reported in *M. gaditana* and *Botryococcus braunii* subjected to 3% CO_2_ supplementation showed increased in the total chlorophyll content and *F_v_*/*F_m_* ratio [32,60]. Further, drastic decrease was found in total chlorophyll, along with reduced (*F_v_*/*F_m_*) and ETRII in ND_HC_ condition demonstrating cellular stress and damaged reaction centres, thus depicting drastic decline in photosynthetic activity. Zao et al. [61] reported similar results in the *Porphyridium cruentum* subjected to nitrogen deplete condition where the photochemical efficiency of PSII decreased along with photoinhibition, indicating the phenomenon of nutrient stress. Further, NL_HC_ supplemented cells depicted slight decrease in (*F_v_*/*F_m_*) and ETRII due to non-availability of nitrate in medium on the 10th day of cultivation (Figure 1).

Furthermore, the DHE fluorescent probes were used to visualize superoxide (O_2_^−^) formation, which is a major damaging ROS in green photosynthetic organisms [62]. ND_HC_ supplemented cells demonstrate higher DHEox fluorescence followed by ND_VLC_ and NL_HC_ conditions, suggesting higher accumulation of ROS radicals (Figure 3A,B). Additionally, only trace amounts of superoxide were accumulated in NR_VLC_ and NR_HC_, depicting lower oxidative stress. Reactive oxygen species (O_2_^−^) are produced in the thylakoid membrane when the absorption of light by chlorophylls exceeds the capacity for energy consumption by photosynthetic apparatus. They play a key role in cell defense, cell signaling, and apoptosis [63]. However, the over accumulation may cause oxidative damage to cellular components i.e., proteins, lipids, and nucleic acids. Moreover, to prevent the harmful effects of ROS, antioxidant defense system like tocopherols and carotenoids play an significant role in scavenging these free radicals [64]. Similarly, ND_HC_ give rise to the highest level of total tocopherol content (2743.03 ± 7.21 µg.g^−1^ DCW) followed by NL_HC_ (1879.38 ± 137.51 µg.g^−1^ DCW) and ND_VLC_ (1393.16 ± 32.2 µg.g^−1^ DCW) conditions, respectively.

Our results show higher total tocopherol productivity in NL_HC_ (734.38 ± 11.79 µg. L^−1^ d^−1^) without compromising growth. Studies has demonstrated that the tocopherols help in protecting the oxidative modification of D1 protein in *Arabidopsis* sp. Further, the tocopherol-deficient mutant (*vte1*) D1 protein was modified to either tyrosine hydroperoxide or dihydroxyphenylalanine, suggesting the role of tocopherols in protecting PSII from oxidative damage [39,65]. Moreover, tocopherols compensate for the loss of xanthophyll cycle pigments (zeaxanthin), exhibiting overlapping function during the photooxidative stress [14] and the *npq1* mutant of *A. thaliana*, accumulates more α-tocopherol in early leaves when exposed to high light [15]. Increased zeaxanthin in *Arabidopsis vte1* mutant, which is tocopherol-deficient, suggests their functional interaction against lipid peroxidation and photooxidative stress [16]. Interestingly, we found similar findings in the stress conditions like ND_VLC_, ND_HC_, and NL_HC_ conditions wherein decrease of zeaxanthin content was observed with improved total tocopherol yields. Moreover, total antioxidant activity estimated by TAC, FRAP, and DPPH assays showed higher antioxidant efficiency in NR_HC_ followed by NL_HC_ and NR_VLC_ conditions, respectively (Table 5). Further, N-limitation decreased the total antioxidant activity in *Phaeodactylum tricornutum*, *Tetraselmis suecica* and *C. vulgaris* suggesting the synergistic effect of different antioxidants to enhance the overall antioxidant activity within the cells [66].

In addition, both total tocopherols and lipids are induced to accumulate under stress conditions like ND_HC_ and NL_HC_ in *Monoraphidium* sp. CABeR41. Specifically, to depict the corelation between these two varaiables from different time points in the specific condition, a linear regression model with *R*^2^ > 0.90 (Appendix A) was observed. Similar reports were illustrated by Chen et al. [67] and Liu et al. [68] depicting relationship between astaxanthin and lipid biosynthesis in *Haematococcus pluvialis* and *C. zofingiensis.* Qualitative (untargeted) metabolomics was done to identify molecular changes in the *Monoraphidium* sp. CABeR41 in excess carbon and nitrogen limitation conditions, utilizing a feedbackward omics approach to elucidate metabolic pathways involved in tocopherol biosynthesis. A total number of nearly 50 metabolites were analyzed and normalized with ribitol as an internal standard [69,70].

In NL_HC_ condition, the metabolites such as glycine was downregulated than the NR_VLC_ cells, which depicts lower photorespiration activity in presence of 3% CO_2_ (*v*/*v*) supplementation. Glycine is components of the photorespiratory glycolate cycle in algae, and its overexpression is used to assess photorespiratory activity [71]. Competition between O_2_ and CO_2_ depicts the rate of carbon absorption, photosynthetic efficiency along with reduced quotient and net photosynthesis [72]. In addition, sugar molecules derived from polysaccharide breakdown are often, increased in nutrient limiting conditions, indicating a reduction in stored carbohydrates to supply a carbon skeleton for vital metabolic processes. Two metabolites, trehalose (1.5-fold) and mannose (3.47-fold) were shown to be elevated under NL_HC_ condition. Trehalose, a non-reducing disaccharide, recognized to be a stabilizing agent for membranes stability from damage aids in the preservation of cellular integrity [69]. Moreover, the accumulation of mannose acts as ROS scavenger during the initial oxidative stress [73]. Further, an increase in mannose has been observed through the breakdown of polysaccharides and glycoproteins in *Isochrysis galbana* during nutrient limited condition [74]. Glutamic acid, an amino acid was also found to upregulate and considered as a vital intermediate for protein synthesis [75,76,77]. It has been reported that glutamic acid accumulates in *P. tricornutum* and *Isochrysis zhangjiangensis* during early stages of nitrogen deficiency, ensuring nitrate assimilation, and providing required nitrogen for protein synthesis [78,79].

Also, significant increase in gamma-aminobutyric acid (GABA) (2.75-fold), a glutamate derivative was observed in the NL_HC_ supplemented cells. GABA, a non-protein amino acid, quickly accumulates in photosynthetic organisms in response to biotic and abiotic stress and governs endogenous signaling and cell development [80]. Increases in glutamate during the early stages of culture under nitrogen limitation may indicate amino acid catabolism, which may be converted to GABA by a decarboxylase process, resulting in nitrogen storage in the form of GABA. These findings suggest that under NL_HC_ conditions, protein and amino acid accumulation was triggered in order to maintain crucial metabolic activities for fast development. Interestingly, another metabolites *myo*-inositol was upregulated (1.41-fold) in NL_HC_ condition, an important component of structural lipids, serves as a phosphate reserve and involved in cell development [81]. An exogenous supply of *myo*-inositol (500 mg L^−1^) to *Dunaliella salina* increased biomass up to 1.34-fold with a rise in lipid productivity and similar comparable results were also reported in *Monoraphidium* sp. QLY-1 [82,83].

Meanwhile, malic acid an intermediate in tricarboxylic acid (TCA) cycle, was also significantly upregulated (1.24-fold), which catalyzes the oxidative decarboxylation of malate to pyruvate by converting NADP to NADPH, thus providing a supply of NADPH for fatty acid biosynthesis [84]. In a recent study, exogenous malic acid supplementation enhanced DHA by 47% in *Schizochytrium* sp. B4D1 [85]. Also, the biochemical profiles of *Monoraphidium* sp. CABeR41, revealed that lipid synthesis rises in NL_HC_, which strongly co-related with our metabolomic profiling. Another TCA cycle intermediate, fumaric acid was found to be upregulated (6.15-fold), involved in decarboxylation of metabolites acting as carbon sink similar to starch which metabolizes to release energy and carbon skeleton for the production of other compounds [86,87]. A study on *pgm1* mutant of *Arabidopsis* sp. which lacks starch, accumulated more fumarate which appears to act as both a transient carbon sink for photosynthate and a pH regulator in nitrate absorption [88]. Therefore, the upregulation of TCA processes may imply towards increased availability of carbon skeletons and reducing power for fatty acids and HVAB biosynthesis [89].

Moreover, downregulation of lactic acid (1.06-fold) signifies the shift of carbon flux towards HVABs. The phenomenon is also supported by elevated tocopherol by nearly 3.25-folds in NL_HC_ supplemented cells. Tocopherols are synthesized by the combination of two major pathways: the MEP (methylerytrithol phosphate pathway) route and the shikimate pathway. The primary structure of tocopherol incorporates polar chromanol head group attached with hydrophobic prenyl side chain synthesized from the farnesyl diphosphate (FPP). The chromanol ring is provided by the shikimate route, whereas the hydrophobic prenyl tail is provided by the MEP pathway [90,91]. Moreover, it competes with carotenoid and squalene biosynthesis in the MEP route because of same precursor, FPP (Figure 6). Our data analysis depicts that total carotenoids as well as squalene accumulation was downregulated in NL_HC_, thus specifying the carbon flux towards tocopherol biosynthesis. Apart from de novo synthesis, tocopherols are synthesized from chlorophyll degradation, where released free phytol is phosphorylated to phytylmonophosphate (phytyl-P) and phytyl-PP by phytol kinase (VTE5) and is fed into biosynthesis of tocopherols, resulting in upregulation of tocopherol during cellular stress [92]. Thus, our schematic representing in Figure 6, depicts a crosstalk between essential metabolites such as glycine, threlose, mannose, malic acid, fumaric acid, GABA, and phytol for concomitant increase of biomass with other biocommodities, namely lipids and tocopherols, in the native isolate *Monoraphidium* sp. CABeR41.

## 5. Conclusions

In conclusion, the new indigenous microalgae *Monoraphidium* sp. CABeR41 when subjected to NL_HC_ condition demonstrates a concomitant increase in cell biomass (316.63 ± 15.47 mg·L^−1^ d^−1^), total lipids (76.12 ± 13.39 mg·L^−1^ d^−1^) and total tocopherols (734.38 ± 11.79 µg L^−1^ d^−1^), respectively. Furthermore, such stratergy demonstrates the cost-effective method for accumulating increased levels of total tocopherols and lipids employing carbon supplementation with varying nitrate concentrations leading to multi-fold enhancement of biocommodities without compromising growth. Also, illustrates the relevance of carbon and nitrogen (C:N) ratio which plays a significant role in achieving higher biomass productivities. In near future, our focus is to employ multi-omics approach to unveil the regulatory metabolic pathways involved in the tocopherol biosynthesis in these photosynthetic cell factories. Perhaps the study is a pivotal laboratory model for scale-up using cost-effective substrates. Such an approach will aim towards fulfilment of sustainable development goals (SDGs) involved in the innovation and cost-effective industrial algal biorefineries providing a better sustainability process with biorenewables, paving a new perspective for the improvement of the bioeconomy.

## Figures and Tables

**Figure 1 cells-11-01315-f001:**
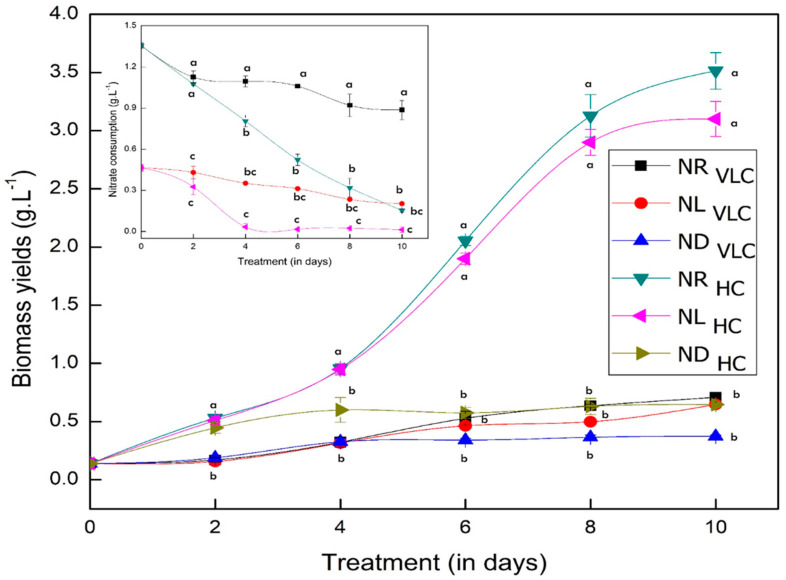
Line diagram indicating biomass yields (in g·L^−1^) of the native isolate *Monoraphidium* sp. CABeR41 subjected to NR_VLC_, NL_VLC_, ND_VLC_, NR_HC_, NL_HC_, and ND_HC_ conditions with nitrate consumption rates (represented as inset in g·L^−1^). Values are the mean average (*n* = 3) ± S.E.; different lowercase letters indicate the statistical significance by two-way ANOVA with *p*-value < 0.05 using *post hoc* analysis by Tukey’s honestly significant difference (HSD).

**Figure 2 cells-11-01315-f002:**
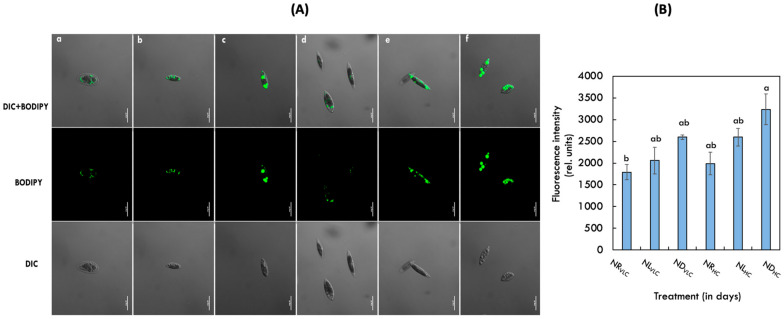
(**A**) BODIPY visualization of *Monoraphidium* cells on the 10th day of cultivation subjected to (**a**) NR_VLC_; (**b**) NL_VLC_; (**c**) ND_VLC_; (**d**) NR_HC_; (**e**) NL_HC_; (**f**) ND_HC_ conditions using confocal microscopy. (**B**) Relative fluorescence intensity of the stained cells on the 10th day of cultivation in the NR_VLC_, NL_VLC_, ND_VLC_, NR_HC_, NL_HC_, and ND_HC_ conditions. Values indicated are mean average (*n* = 3) ± S.E.; different lowercase letters indicate the statistical significance by two-way ANOVA with *p*-value < 0.05 using *post hoc* analysis by Tukey’s honestly significant difference (HSD).

**Figure 3 cells-11-01315-f003:**
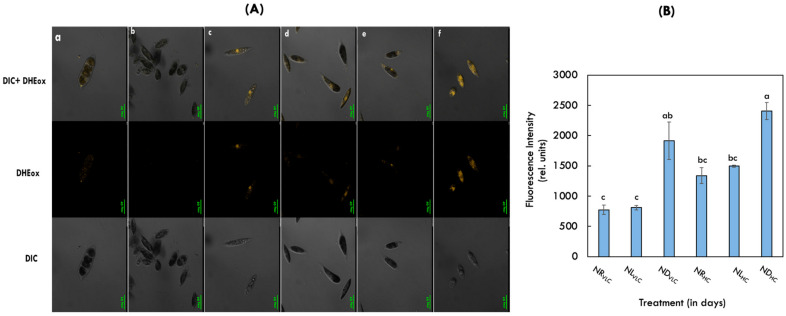
(**A**) Imaging of the Superoxide anion radical activity in the native isolate *Monoraphidium* sp. CABeR41 on the 10th day of cultivation by laser confocal scanning microscopy with an excitation wavelength of 520 nm and emission wavelength of 570 nm subjected to (**a**) NR_VLC_; (**b**) NL_VLC_; (**c**) ND_VLC_; (**d**) NR_HC_; (**e**) NL_HC_; (**f**) ND_HC_ conditions using DHE staining. (**B**) Relative fluorescence intensity of the DHE stained cells on the 10th day of cultivation in the NR_VLC_, NL_VLC_, ND_VLC_, NR_HC_, NL_HC_, and ND_HC_ conditions. Values indicated are mean average (*n* = 3) ± S.E.; different lowercase letters indicate the statistical significance by two-way ANOVA with *p*-value < 0.05 using *post hoc* analysis by Tukey’s honestly significant difference (HSD).

**Figure 4 cells-11-01315-f004:**
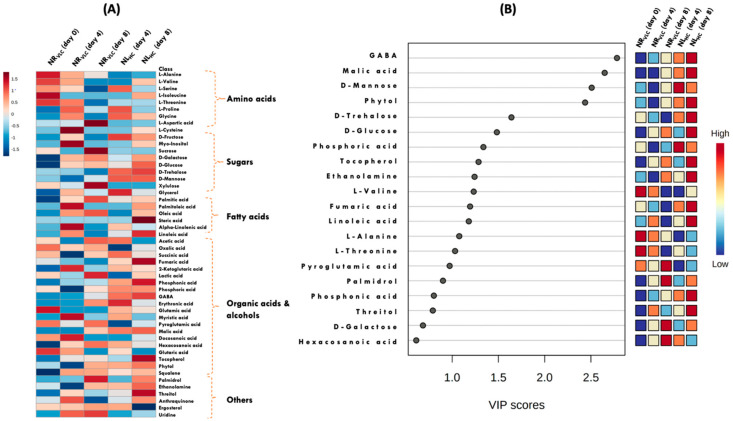
(**A**) Heat map depicting the time-course qualitative (untargeted) metabolomics on the 4th and 8th days of the native isolate *Monoraphidium* sp. CABeR41 subjected to NR_VLC_ and NL_HC_ conditions. (**B**) Variable Importance in Projection (VIP) score plot representing the significant metabolites subjected to NR_VLC_ and NL_HC_ conditions.

**Figure 5 cells-11-01315-f005:**
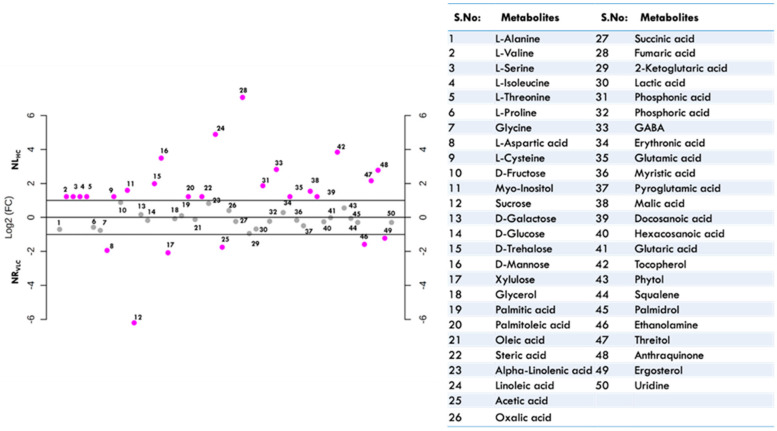
Dot-plot representing metabolites upregulated and downregulated in the native isolate *Monoraphidium* sp. CABeR41 subjected to NR_VLC_ vs. NL_HC_ conditions on the 8th day of cultivation with a summary of table representing the identified metabolites (pink dots represent the metabolites that show >log 2-fold change and grey dots represents the one that show <log 2-fold change).

**Figure 6 cells-11-01315-f006:**
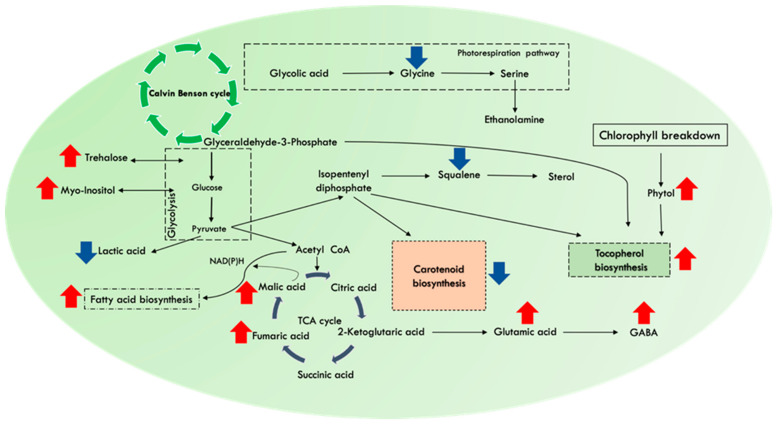
Schematic representation of essential pathways and their metabolites involved in the multi-fold enhancement of tocopherol yields in the native isolate *Monoraphidium* sp. CABeR41 subjected to effect of high CO_2_ supplementation and nitrate limitation (NL_HC_) condition. Upregulation represented in the red colour (↑); downregulation represented in the blue colour (↓).

**Table 1 cells-11-01315-t001:** Comparison of various parameters on the 10th day of cultivation, i.e., biomass yields, productivities, and photosynthetic efficiency subjected to NR_VLC_, NL_VLC_, ND_VLC_, NR_HC_, NL_HC_, and ND_HC_ conditions in the native isolate *Monoraphidium* sp. CABeR41. Values indicated are mean average (*n* = 3) ± S.E.; different lowercase letters indicate the statistical significance by two-way ANOVA with *p*-value < 0.05 using *post hoc* analysis by Tukey’s honestly significant difference (HSD).

Parameters	NR_VLC_	NL_VLC_	ND_VLC_	NR_HC_	NL_HC_	ND_HC_
Biomass (g·L^−1^)	0.71 ± 0.01 ^b^	0.65 ± 0.01 ^b^	0.38 ± 0.01 ^b^	3.54 ± 0.13 ^a^	3.30 ± 0.16 ^a^	0.62 ± 0.03 ^b^
Biomass Productivity (mg·L^−1^ d^−1^)	56.93 ± 0.76 ^b^	50.75 ± 0.89 ^b^	23.39 ± 1.14 ^b^	336.35 ± 14.92 ^a^	316.63 ± 15.58 ^a^	48.19 ± 2.96 ^b^
Specific Growth Rate (µ)	0.25 ± 0.02	0.19 ± 0.06	0.06 ± 0.03	0.44 ± 0.02	0.45 ± 0.01	0.12 ± 0.04
Doubling Time (days)	2.82 ± 0.18	3.58 ± 0.23	11.15 ± 0.86	1.60 ± 0.09	1.54 ± 0.04	6.50 ± 0.02
*F_v_*/*F_m_*	0.78 ± 0.01	0.55 ± 0.03	0.30 ± 0.09	0.80 ± 0.02	0.65 ± 0.06	0.00 ± 0.00
*F_q_′*/*F_m_′*	0.45 ± 0.05	0.29 ± 0.07	0.15 ± 0.04	0.51 ± 0.05	0.36 ± 0.05	0.00 ± 0.00
ETRII	35.90 ± 3.54	23.35 ± 5.62	12.20 ± 3.46	41.10 ± 4.38	29.70 ± 5.16	0.00 ± 0.00
Total Chlorophylls (mg·g^−1^ DCW)	21.35 ± 1.92 ^b^	16.16 ± 1.36 ^b^	3.20 ± 0.54 ^cd^	42.53 ± 3.44 ^a^	15.02 ± 1.58 ^bc^	0.77 ± 0.18 ^d^

µ: specific growth rate; *F_v_*/*F_m_*: Maximum quantum efficiency of PSII photochemistry; *F_q_′*/*F_m_′*: PSII operating efficiency; ETRII: electron transport rate of the PSII reaction centres; DCW: dry cell weight.

**Table 2 cells-11-01315-t002:** Biochemical constituents on the 10th day of cultivation i.e., total proteins, total carbohydrates, and total lipids subjected to NR_VLC_, NL_VLC_, ND_VLC_, NR_HC_, NL_HC_, and ND_HC_ conditions in the native isolate *Monoraphidium* sp. CABeR41. Values indicated are mean average (*n* = 3) ± S.E.; different lowercase letters indicate the statistical significance by two-way ANOVA with *p*-value < 0.05 using *post hoc* analysis by Tukey’s honestly significant difference (HSD).

Biochemical Constitutents	NR_VLC_	NL_VLC_	ND_VLC_	NR_HC_	NL_HC_	ND_HC_
	Yields/Productivities (mg·g^−1^ DCW/mg·L^−1^ d^−1^)					
Total Proteins	508.76 ± 33.6/ 21.37 ± 1.7 ^c^	400.00 ± 21.0/ 16.73 ± 0.2 ^cd^	334.79 ± 67.6/ 0.53 ± 0.53 ^d^	660.43 ± 24.7/ 197.88 ± 6.5 ^a^	353.02 ± 34.1/ 79.47 ± 5.3 ^b^	201.70 ± 46.7/ 1.40 ± 0.5 ^d^
Total Carbohydrates	95.70 ± 3.4/ 5.24 ± 0.2 ^c^	70.37 ± 4.1/ 3.24 ± 0.3 ^c^	205.37 ± 10.4/ 6.10 ± 0.3 ^c^	106.64 ± 4.8/ 35.80 ± 1.3 ^b^	269.41 ± 26.5/ 87.12 ± 7.0 ^a^	177.15 ± 5.1/ 9.87 ± 0.5 ^c^
Total Lipids	114.00 ± 11.8/ 6.08 ± 0.3 ^c^	125.10 ± 21.0/ 7.50 ± 1.2 ^c^	193.15 ± 16.0/ 5.97 ± 0.3 ^c^	169.11 ± 17.5/ 49.68 ± 6.9 ^ab^	214.70 ± 29.8/ 76.12 ± 13.4 ^a^	344.70 ± 16.8/ 17.60 ± 3.3 ^bc^

DCW: dry cell weight.

**Table 3 cells-11-01315-t003:** Summary of tocopherol yields and productivities on the 10th day of cultiva-tion subjected to NR_VLC_, NL_VLC_, ND_VLC_, NR_HC_, NL_HC_, and ND_HC_ conditions in the native isolate *Monoraphidium* sp. CABeR41. Values indicated are mean average (*n* = 3) ± S.E.; different lowercase letters indicate the statistical significance by two-way ANOVA with *p*-value < 0.05 using *post hoc* analysis by Tukey’s honestly significant difference (HSD).

Tocopherols	NR_VLC_	NL_VLC_	ND_VLC_	NR_HC_	NL_HC_	ND_HC_
	Yields/Productivities (μg.g^−1^ DCW/μg·L^−1^ d^−1^)					
α-Tocopherol	539.95 ± 69.2 ^bc^/ 32.93 ± 4.9	425.01 ± 53.0 ^c^/ 30.65 ± 2.9	468.20 ± 48.2 ^bc^/ 19.45 ± 3.5	388.25 ± 3.9 ^c^/ 137.07 ± 3.3	736.76 ± 22.5 ^ab^/ 244.73 ± 7.4	947.72 ± 49.2 ^a^/ 54.85 ± 6.5
δ-Tocopherol	528.10 ± 61.1 ^c^/ 27.42 ± 2.4	687.50 ± 48.3 ^c^/ 30.66 ± 3.1	925.50 ± 47.5 ^c^/ 19.47 ± 0.7	478.20 ± 22.7 ^c^/ 169.07 ± 0.4	1142.63 ± 159.6 ^b^/ 489.65 ± 4.5	1795.31 ± 56.3 ^a^/ 107.01 ± 5.4
Total Tocopherols	1068.04 ± 130.2/ 60.35 ± 7.3 ^de^	1111.51 ± 88.2/ 70.31 ± 8.2 ^d^	1393.16 ± 32.2/ 38.92 ± 4.2 ^e^	866.45 ± 26.7/ 306.14 ± 2.9 ^b^	1879.38 ± 137.5/ 734.38 ± 11.8 ^a^	2743.03 ± 7.21/ 161.85 ± 11.9 ^c^

DCW: dry cell weight.

**Table 4 cells-11-01315-t004:** Estimation of carotenoid yields (in mg·g^−1^ DCW) on the 10th day subjected to NR_VLC_, NL_VLC_, ND_VLC_, NR_HC_, NL_HC_, and ND_HC_ conditions in the native isolate *Monoraphidium* sp. CABeR41. Values indicated are mean average (*n* = 3) ±S.E.; different lowercase letters indicate the statistical significance by two-way ANOVA with *p*-value < 0.05 using *post hoc* analysis by Tukey’s honestly significant difference (HSD).

Carotenoids (in mg·g^−1^ DCW)	NR_VLC_	NL_VLC_	ND_VLC_	NR_HC_	NL_HC_	ND_HC_
Lycopene	0.08 ± 0.02 ^a^	0.05 ± 0.01 ^a^	0.17 ± 0.03 ^a^	0.01 ± 0.00 ^a^	0.09 ± 0.00 ^a^	0.14 ± 0.00 ^a^
α-Carotene	0.22 ± 0.05 ^b^	0.19 ± 0.04 ^bc^	0.01 ± 0.00 ^c^	0.72 ± 0.02 ^a^	0.07 ± 0.02 ^bc^	0.00 ± 0.01 ^c^
β-carotene	0.30 ± 0.07 ^b^	0.20 ± 0.00 ^b^	0.11 ± 0.02 ^a^	0.49 ± 0.10 ^b^	0.22 ± 0.07 ^a^	0.03 ± 0.02 ^a^
Zeaxanthin	0.91 ± 0.05 ^b^	0.75 ± 0.07 ^bc^	0.45 ± 0.01 ^d^	1.43 ± 0.07 ^a^	0.62 ± 0.01 ^bc^	0.20 ± 0.01 ^c^
Echinenone	0.03 ± 0.01 ^b^	0.01 ± 0.00 ^b^	0.32 ± 0.03 ^a^	0.02 ± 0.00 ^b^	0.21 ± 0.04 ^a^	0.35 ± 0.07 ^a^
Total carotenoids	1.54 ± 0.08 ^b^	1.2 ± 0.05 ^bc^	1.06 ± 0.09 ^bc^	2.67 ± 0.15 ^a^	1.21 ± 0.21 ^bc^	0.72 ± 0.08 ^c^

DCW: dry cell weight.

**Table 5 cells-11-01315-t005:** Summary of antioxidant activities on the 10th day performed by TAC, FRAP, and DPPH assays (represented as mg·g−1 ascorbic acid equivalent) subjected to NR_VLC_, NL_VLC_, ND_VLC_, NR_HC_, NL_HC_, and ND_HC_ conditions in the native isolate *Monoraphidium* sp. CABeR41. Values indicated are mean average (*n* = 3) ± S.E.; different lowercase letters indicate the statistical significance by two-way ANOVA with *p*-value < 0.05 using *post hoc* analysis by Tukey’s honestly significant difference (HSD).

Assays (mg·g^−^^1^)	NR_VLC_	NL_VLC_	ND_VLC_	NR_HC_	NL_HC_	ND_HC_
TAC	24.77 ± 4.02 ^a^	28.23 ± 4.08 ^a^	15.94 ± 1.88 ^b^	38.60 ± 9.23 ^a^	28.96 ± 7.32 ^a^	12.31 ± 2.17 ^b^
FRAP	11.08 ± 2.65 ^a^	10.13 ± 2.30 ^a^	10.70 ± 3.36 ^a^	13.30 ± 0.56 ^a^	13.36 ± 3.29 ^a^	5.07 ± 2.46 ^b^
DPPH	11.03 ± 4.77 ^b^	13.43 ± 6.57 ^b^	10.26 ± 9.75 ^c^	19.96 ± 5.00 ^a^	15.83 ± 4.30 ^b^	14.90 ± 4.28 ^b^

## Data Availability

Data is contained within the article or Appendix A.

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
