# Peer review of "Multi-Fold Enhancement of Tocopherol Yields Employing High CO_2_ Supplementation and Nitrate Limitation in Native Isolate *Monoraphidium* sp."

_cells, 2022, doi:10.3390/cells11081315_

Round 1

Reviewer 1 Report

The article “Multi-Fold Enhancement of Tocopherol Yields Employing Synergistic Effect of High CO2 Supplementation and Nitrate Limitation in Native Isolate Monoraphidium sp.” analyzes the biotechnological potential of a novel algae isolate by studying the influence of CO2 addition and N limitation.

The manuscript is well written, the quality of the quality of the methodology is great and the article certainly deserves considering publication, which is what I would recommend with the following changes.

Major points

The authors should test for significance for most of the work, it sometimes is stated in the text that findings were significant, but it is not described how these were obtained for the specific experiment. The methods just describe a general approach, which is fine at that particular part of the manuscript, but for the figures and tables, the individual significant results should be highlighted, the comparisons that were tested should be indicated, and the individual methodology used for the particular comparison should be mentioned. I do believe the general takeaways of the authors from the data are valid. This is true for the whole manuscript, but especially crucial for Fig 2B, 3B and the metabolite data (Fig 5).

Minor points:

Fig1 inset should have same coloration and symbols as main Fig1. Currently NR(HC) has the same coloration and symbol as NL(HC), which can be confusing for the reader when looking at the figure briefly. A change would also allow to remove the 2nd legend in the inset.

Fig5 the color panel at the top of the heatmap and the legend with colors used therein are unnecessary. Columns are labeled already, the colors (red especially) overlaps with the heatmap, I would suggest to remove

Synergistic effect between CO2 addition and N limitation. I would argue with the validity of this point in the sense that total C fixation is not increased (judged by total biomass (Table1) and NL alone is not increasing lipid or carbohydrate content in low CO2 (Table 1). So individually N limitation is not advancing the production of lipids or carbohydrates, only in the context of improved CO2 supply the changes are happening. This is more a conditional phenotype. I think this could be worded better to avoid confusion.

Author Response

Manuscript ID: Cells-1634160

Title: Multi-Fold Enhancement of Tocopherol Yields Employing High CO2 Supplementation and Nitrate Limitation in Native Isolate Monoraphidium sp.

Response: All the feedbacks and comments provided by all the Reviewer’s/Editor, have been taken care of with the best of our capacity. We thank Reviewer’s for his/her encouraging positive feedback and finding the work interesting. Also, for the critical and constructive comments, helped us to improve the submitted manuscript. We tried to address all the questions raised by Reviewer(s)/Editor. The modifications made in the revised manuscript in response to the comments are highlighted in yellow for the ease in tracking the changes conveniently. We have also included page and line numbers for ease of tracking all the changes done in the revised manuscript.

Point-to-point responses to the Reviewer#1 comments:

Previous title “Multi-Fold Enhancement of Tocopherol Yields Employing Synergistic Effect of High CO2 Supplementation and Nitrate Limitation in Native Isolate Monoraphidium sp.” has been rephrased as follows:

“Multi-Fold Enhancement of Tocopherol Yields Employing High CO2 Supplementation and Nitrate Limitation in Native Isolate Monoraphidium sp.”

  • The authors should test for significance for most of the work, it sometimes is stated in the text that findings were significant, but it is not described how these were obtained for the specific experiment. The methods just describe a general approach, which is fine at that particular part of the manuscript, but for the figures and tables, the individual significant results should be highlighted, the comparisons that were tested should be indicated, and the individual methodology used for the particular comparison should be mentioned. I do believe the general takeaways of the authors from the data are valid. This is true for the whole manuscript, but especially crucial for Fig 2B, 3B and the metabolite data (Fig 5).

Response: We thank Reviewer#1 for his/her encouraging positive feedback and finding the work interesting. As suggested by the reviewer, we have included the following changes in the revise manuscript as follows: the statistical significance by two-way ANOVA with p-value < 0.05 with different lowercase letters are represented in all the Figures and Tables including the Figures 2B, 3B and 4B (a new figure was included in the metabolomics data with VIP scores to demonstrate the significance).

  • Fig1 inset should have same coloration and symbols as main Fig1. Currently NR (HC) has the same coloration and symbol as NL(HC), which can be confusing for the reader when looking at the figure briefly. A change would also allow to remove the 2nd legend in the inset.

Response: As suggested by the reviewer, we have revised the Figure 1 inset with same coloration and symbols as the main Figure 1 [Page no.: 6, Line nos.: 256-281]

  • Fig5 the color panel at the top of the heat map and the legend with colors used therein are unnecessary. Columns are labeled already, the colors (red especially) overlaps with the heat map, I would suggest to remove.

Response: As suggested by the reviewer, we have deleted the color panel from the Figure 5 (now reclassified as Figure 4A in the revised manuscript - Page no. 11, Line nos.: 528-549). Included new plot i.e., which includes VIP scores representing the significant metabolites subjected to NRVLC and NLHC conditions (now included as Figure 4B in the revised manuscript - Page no. 11, Line nos.: 528-549).

  • Synergistic effect between CO2 addition and N limitation. I would argue with the validity of this point in the sense that total C fixation is not increased (judged by total biomass (Table1) and NL alone is not increasing lipid or carbohydrate content in low CO2 (Table 1). So individually N limitation is not advancing the production of lipids or carbohydrates, only in the context of improved CO2 supply the changes are happening. This is more a conditional phenotype. I think this could be worded better to avoid confusion.

Response: We agree with the reviewer’s comment and has rephrased the sentences in the revised manuscript. For example: Synergistic effect has been deleted for better clarity in the revised manuscript.

Reviewer 2 Report

Line 22, an acronym is entered without expanding it (NLHC)
Line 94-97: The stated quantities correspond to the preparation of the BG11 stocks, they state it as if the quantities are the final concentration of the culture medium.
Line 49/380: Unify symbol format for superoxide radicals
Line 617-658: very long paragraphs.
Line 551: very poor conclusion, it does not reflect what is in the summary and with all the information generated the conclusions can be expanded

Author Response

Manuscript ID: Cells-1634160

Title: Multi-Fold Enhancement of Tocopherol Yields Employing High CO2 Supplementation and Nitrate Limitation in Native Isolate Monoraphidium sp.

Response: All the feedbacks and comments provided by all the Reviewer’s/Editor, have been taken care of with the best of our capacity. We thank Reviewer’s for his/her encouraging positive feedback and finding the work interesting. Also, for the critical and constructive comments, helped us to improve the submitted manuscript. We tried to address all the questions raised by Reviewer(s)/Editor. The modifications made in the revised manuscript in response to the comments are highlighted in yellow for the ease in tracking the changes conveniently. We have also included page and line numbers for ease of tracking all the changes done in the revised manuscript.

Point-to-point responses to the Reviewer#2 comments:

Previous title “Multi-Fold Enhancement of Tocopherol Yields Employing Synergistic Effect of High CO2 Supplementation and Nitrate Limitation in Native Isolate Monoraphidium sp.” has been rephrased as follows:

“Multi-Fold Enhancement of Tocopherol Yields Employing High CO2 Supplementation and Nitrate Limitation in Native Isolate Monoraphidium sp.”

  • Line 22, an acronym is entered without expanding it (NLHC).

Response: Thank you for the comment and has included the acronym in the revised manuscript as follows [Abstract, Page no.: 1, Line no.: 22]: NLHC (Nitrate limited + 3% CO2).

  • Line 94-97: The stated quantities correspond to the preparation of the BG11 stocks, they state it as if the quantities are the final concentration of the culture medium.

Response: As suggested by the reviewer, we have included the final concentrations of the culture medium in the revised manuscript as follows [Page no.: 3, Line nos.: 102-107]: “The final concentration of BG-11 medium used in culture conditions are represented in mg.mL-1, mM: K2HPO4 – 40, 0.23; …Co(NO3)2.6H2O - 0.05, 0.005)”.

  • Line 49/380: Unify symbol format for superoxide radicals.

Response: We have included unify symbol format for superoxide radicals (O2) in the revised manuscript as suggested by the reviewer.

  • Line 617-658: very long paragraphs.

Response: As suggested by the reviewer, we have shorten the paragraphs and also rephrased the same in the revised manuscript as follows [Page nos.: 13, 14, Line nos.: 649-663, 664-681]:

“Furthermore, the DHE fluorescent probes were used to visualize superoxide (O2-) formation…conditions, respectively”.

“Our results show higher total tocopherol productivity in NLHC… activity within the cells [66]”.

  • Line 551: very poor conclusion, it does not reflect what is in the summary and with all the information generated the conclusions can be expanded.

Response: As suggested by the reviewer, we have rephrased the conclusion section with our findings and future perspectives as follows in the revised manuscript [Page no. 16, Line nos.: 785-799]: “In conclusion, the new indigenous microalgae Monoraphidium sp. CABeR41 when subjected to NLHC condition…process with biorenewables, paving a new perspective for the improvement of the bioeconomy.”

Reviewer 3 Report

Multi-fold enhancement of tocopherol yields was achieved by combining CO2 supplementation and nitrate availability in this study. The authors also explored the potential molecular mechanism of tocopherol accumulation using global metabolomics. This study in certainly helpful in terms of using high CO2 to produce   high-value added biorenewables with microalgae (HVABs). Meanwhile, there are some points that the authors need to attend to before this paper can be published in Cells.

Specific comments

Line 22 What is NLHC?

Line 27 What are GABA and TCA? Please use full spellings for the first occurrence.

Line 28 Which in ???

Line 43 The authors may cite the latest studies that culture microalgae using high CO2 levels to produce sustainable biofuels and HVABs, such as Wu et al (2022).

Wu, M., Gao, G., Jian, Y. and Xu, J., 2022. High CO2 increases lipid and polyunsaturated fatty acid productivity of the marine diatom Skeletonema costatum in a two-stage model. Journal of Applied Phycology34(1), pp.43-50.

Line 76 Before the last paragraph, the authors should review the studies in regard to the effects of CO2 and nitrogen on tocopherols production from microalgae.

Line 106 I am wondering how 0.03% and 3% CO2 levels were supplied and how CO2 and nitrate levels were maintained during 10 days of culture? 3% CO2 is very high, what is the corresponding pH of medium?

Line 225 Two-way ANOVA rather than one-way should be used to analyze the data since there were two factors in this study.

What does DCW mean in Tables 1 and 2?

Line 332 Delete "Tabulation of".

Lines 359 and 437 The layout of figures and tables is very strange. The legends are sometimes above and sometimes below the figures/tables.

Line 248 I can see that the authors did not maintain the nitrogen levels as set at the beginning. Nitrate was completely used up by day 4 for the treatment of NLHC. Then what is the difference between the two treatments, NLHC and NDHC?

Line 567 For the balance of grow rate and lipid content, the authors may refer to recent studies (Jiang et al., 2016; Gao et al., 2019)
Gao G, Wu M, Fu Q, Li X, Xu J. A two-stage model with nitrogen and silicon limitation enhances lipid productivity and biodiesel features of the marine bloom-forming diatom Skeletonema costatum. Bioresource Technology. 2019, Jun 27:121717.

Jiang XM, Han QX, Gao XZ, and Gao G*. 2016. Conditions optimising on yield of biomass, total lipid, and valuable fatty acids in two strains of Skeletonema menzelii. Food Chemistry. 194: 723-732. 

Line 760 In conclusions, flaws of the current study or future study needs are usually mentioned.

Author Response

Manuscript ID: Cells-1634160

Title: Multi-Fold Enhancement of Tocopherol Yields Employing High CO2 Supplementation and Nitrate Limitation in Native Isolate Monoraphidium sp.

Response: All the feedbacks and comments provided by all the Reviewer’s/Editor, have been taken care of with the best of our capacity. We thank Reviewer’s for his/her encouraging positive feedback and finding the work interesting. Also, for the critical and constructive comments, helped us to improve the submitted manuscript. We tried to address all the questions raised by Reviewer(s)/Editor. The modifications made in the revised manuscript in response to the comments are highlighted in yellow for the ease in tracking the changes conveniently. We have also included page and line numbers for ease of tracking all the changes done in the revised manuscript.

Point-to-point responses to the Reviewer#3 comments:

Previous title “Multi-Fold Enhancement of Tocopherol Yields Employing Synergistic Effect of High CO2 Supplementation and Nitrate Limitation in Native Isolate Monoraphidium sp.” has been rephrased as follows:

“Multi-Fold Enhancement of Tocopherol Yields Employing High CO2 Supplementation and Nitrate Limitation in Native Isolate Monoraphidium sp.”

  • Line 22 What is NLHC?

Response: Thank you for the comment and has included the acronym in the revised manuscript as follows [Abstract, Page no.: 1, Line no.: 22]: NLHC (Nitrate limited + 3% CO2).

  • Line 27 What are GABA and TCA? Please use full spellings for the first occurrence.

Response: We have included the acronym in the revised manuscript as follows [Abstract, Page no.: 1, Line no.: 27, 28]: gamma-Aminobutyric acid (GABA) and tricarboxylic acid (TCA).

  • Which in?

Response: We have revised the same as follows [Abstract, Page no.: 1, Line no.: 29-30]: ‘which is leading to increased biomass yields along with the other biocommodities’.

  • Line 43 The authors may cite the latest studies that culture microalgae using high CO2 levels to produce sustainable biofuels and HVABs, such as Wu et al (2022).

Wu, M., Gao, G., Jian, Y. and Xu, J., 2022. High CO2 increases lipid and polyunsaturated fatty acid productivity of the marine diatom Skeletonema costatum in a two-stage model. Journal of Applied Phycology34(1), pp.43-50.

Response: As suggested by the reviewer, we have included the following reference i.e., Wu et al (2022) in the revised manuscript [Page no.: 1, Line no.: 43] as follows: “converting it into sustainable biofuels and high-value added biorenewables (HVABs) [1-5]”.  

  • Line 76 Before the last paragraph, the authors should review the studies in regard to the effects of CO2 and nitrogen on tocopherols production from microalgae.

Response: As suggested by the reviewer, we have included previous studies supporting our study in the revised manuscript [Page no.: 2, Line nos.: 75-82] as follows: “In addition, previous studies involving strains such as Coccomyxa sp., Desmodesmus sp., and Muriella terrestris supplemented with 5% CO2 (v/v)… the influence of nitrate limited condition [30]”.

  • Line 106, I am wondering how 0.03% and 3% CO2 levels were supplied and how CO2 and nitrate levels were maintained during 10 days of culture? 3% CO2 is very high, what is the corresponding pH of medium?

Response: Thanks for the comment and agree with the reviewer. In the present study, we measured the pH at regular time intervals and has observed that pH ranges between 7.5-8.25 in all the conditions when supplemented with CO2, which is an ideal growth condition for the microalgae.

  • Line 225 Two-way ANOVA rather than one-way should be used to analyze the data since there were two factors in this study.

Response: We agree with the reviewer and has included the statistical significance by two-way ANOVA with p-value < 0.05 with different lowercase letters are represented in all the Figures and Tables in the revised manuscript (highlighted in the text in yellow).

  • What does DCW mean in Tables 1 and 2?

Response: We have included the full abbreviation of ‘DCW’ in the revised manuscript as follows: “DCW: dry cell weight”.

  • Line 332 Delete "Tabulation of".

Response: Deleted the ‘Tabulation of’ from the Table 2 as suggested by the reviewer [Page no.: 8, Line no.: 358].

  • Lines 359 and 437 The layout of figures and tables is very strange. The legends are sometimes above and sometimes below the figures/tables.

Response: As suggested by the reviewer, we have included the figure and table legends (uniformly for better clarity) in the revised manuscript.

  • Line 248 I can see that the authors did not maintain the nitrogen levels as set at the beginning. Nitrate was completely used up by day 4 for the treatment of NLHC. Then what is the difference between the two treatments, NLHC and NDHC?

Response: We agree with the reviewer’s comment and would like to justify that the NLHC (i.e., 0.5 g.L-1) condition is one-quarter of the concentration used in BG-11 medium (i.e., NRHC 1.5 g.L-1) whereas in NDHC (no nitrate is present i.e., i.e., 0.0 g.L-1). Henceforth, the concentrations in all the above conditions vary at the beginning.

  • Line 567 For the balance of grow rate and lipid content, the authors may refer to recent studies (Jiang et al., 2016; Gao et al., 2019)

Gao G, Wu M, Fu Q, Li X, Xu J. A two-stage model with nitrogen and silicon limitation enhances lipid productivity and biodiesel features of the marine bloom-forming diatom Skeletonema costatum. Bioresource Technology. 2019, Jun 27:121717.

Jiang XM, Han QX, Gao XZ, and Gao G*. 2016. Conditions optimising on yield of biomass, total lipid, and valuable fatty acids in two strains of Skeletonema menzelii. Food Chemistry. 194: 723-732.

Response: We have included the following references in the revised manuscript as follows [Page no.: 12, Line nos.: 599] as suggested by the reviewer.

  • Line 760 In conclusions, flaws of the current study or future study needs are usually mentioned.

Response: As suggested by the reviewer, we have rephrased the conclusion section with our findings and future perspectives as follows in the revised manuscript [Page no. 16, Line nos.: 785-799]: “In conclusion, the new indigenous microalgae Monoraphidium sp. CABeR41 when subjected to NLHC condition…process with biorenewables, paving a new perspective for the improvement of the bioeconomy.”

Round 2

Reviewer 3 Report

The authors carefully revised the manuscript based on my comments. I do not have any further comment.

Author Response

We thank Reviewer for his/her encouraging positive feedback and finding the work interesting.